# Novel Micellar Formulation of Silymarin (Milk Thistle) with Enhanced Bioavailability in a Double-Blind, Randomized, Crossover Human Trial

**DOI:** 10.3390/pharmaceutics17070880

**Published:** 2025-07-04

**Authors:** Chuck Chang, Yiming Zhang, Yun Chai Kuo, Min Du, Kyle Roh, Roland Gahler, Afoke Ibi, Julia Solnier

**Affiliations:** 1ISURA, Burnaby, BC V3N4S9, Canada; cchang@isura.ca (C.C.); yzhang@isura.ca (Y.Z.); mdu@isura.ca (M.D.); kroh@isura.ca (K.R.); aibi@isura.ca (A.I.); 2Factors Group R & D, Burnaby, BC V3N4S9, Canada

**Keywords:** milk thistle, silymarin, micelles, LipoMicel, bioavailability, pharmacokinetics, liver health, human study, hepatoprotective

## Abstract

**Background:** Silymarin, a flavonoid complex, and the main bioactive component of milk thistle (*Silybum marianum*), is known for its hepatoprotective properties but suffers from poor bioavailability due to its low solubility and extensive first-pass metabolism. **Method:** This study aimed to evaluate the pharmacokinetics and tolerability of a novel micellar milk thistle formulation designed to enhance silymarin absorption, compared to an unformulated/standard milk thistle product, in a small-scale human bioavailability trial. In a randomized, double-blinded, crossover study, 16 healthy participants received a single dose of either the micellar formulation (LipoMicel Milk Thistle; LMM) or the standard formulation (STD) at a total daily dose of 130 mg silymarin. Blood concentrations were measured over 24 h, and key pharmacokinetic parameters—maximum plasma concentration (C_max_), time to reach maximum concentration (T_max_), and area under the curve (AUC)—were calculated. Tolerability and safety were assessed through adverse event monitoring during the study period. **Results:** Results demonstrated a significant increase in bioavailability with the micellar formulation, with 18.9-fold higher C_max_ (95% CI: 1.9–30.7 ng/mL vs. 74.4–288.3 ng/mL; *p* = 0.007) and 11.4-fold higher AUC_0–24_ (95% CI: 7.40–113.5 ng·h/mL vs. 178–612.5 ng·h/mL; *p* = 0.015). T_max_ was 0.5 (95% CI: 0.5–4.0) hours for the micellar formulation versus 2.5 (95% CI: 0.5–8.0) hours for the standard product (*p* = 0.015) indicating faster absorption of LMM. The standard formulation exhibited a significantly longer mean residence time compared to the LMM formulation (95% CI: 4.4–7.5 h vs. 2.8–4.2 h; *p* = 0.015). **Conclusions:** No adverse events or significant safety concerns were observed in either group. Compared to the standard, the micellar formulation showed superior pharmacokinetic outcomes, suggesting it may enhance silymarin’s clinical efficacy in liver health.

## 1. Introduction

Silymarin is a mixture of structurally similar flavonoid compounds primarily consisting of silybin, isosilybin, silydianin, silychristin, and isosilychristin (Figure 1). It is the principal active ingredient in the milk thistle plant (*Silybum marianum*) and has been extensively studied for its hepatoprotective and antioxidant properties [1,2,3,4]. In fact, silymarin (Milk Thistle) supplementation has been shown to effectively lower important clinical parameters such as liver enzyme levels [5,6]. Therefore, it has been studied as a potential treatment for diseases including alcoholic liver disease, non-alcoholic fatty liver disease, and viral hepatitis, among others [3,4]. And, as with most flavonoids, silymarin has been studied for its impact on inflammation and oxidative stress [7,8,9]. Moreover, several studies reported positive results on the use of silymarin in treating cancer [10,11,12]. However, despite its promising pharmacological potential, the clinical use of silymarin and its primary component silybin has been limited by its poor bioavailability [2,13,14]. The lipophilic nature of these compounds results in low solubility in water, and once absorbed, extensive first-pass metabolism further reduces its therapeutic efficacy when administered in standard formulations [2]. To address these limitations, numerous methods have been developed to enhance the bioavailability of silymarin, including novel formulations such as liposomes, effervescent tablets, nanocrystals, co-crystallization, and self-microemulsifying delivery systems (SMEDS) [15,16,17,18,19,20,21].

One such promising advancement is the use of micellar formulations, which have been shown to improve the solubility and membrane permeability of hydrophobic compounds [22,23,24,25]. Micelles are colloidal aggregates that can encapsulate hydrophobic molecules, protecting them from degradation and promoting their transport across biological membranes. Formulations containing therapeutic micelles have gained attention in pharmaceutical research due to their potential to significantly enhance the bioavailability of various compounds, including silymarin [14,26,27,28]. Given the importance of liver function in conditions such as chronic liver disease and the increasing use of complementary and alternative medicines, enhancing the efficacy of silymarin through advanced drug delivery systems could offer significant therapeutic advantages [29,30,31].

In this study, LipoMicel^®^ silymarin formulation (LMM) represents a novel, commercially available formulation that utilizes an anhydrous self-emulsifying system to promote in situ micelle formation upon ingestion, thereby enhancing solubility and absorption without requiring aqueous pre-dispersion. This study aims to evaluate the pharmacokinetics and potential therapeutic benefits of LMM compared to its unformulated (standard, STD) counterpart in human participants. While the enhancement of bioavailability through micellar delivery systems has been demonstrated in vitro and in animal models, human clinical data substantiating these findings are limited [13,32,33,34,35,36]. Therefore, a small-scale clinical trial was conducted to compare the blood concentrations of total silymarin after oral ingestion. Specifically, this study focuses on key pharmacokinetic parameters such as maximum plasma concentration (C_max_), time to reach maximum concentration (T_max_), and area under the plasma concentration-time curve (AUC), which are critical for understanding the bioavailability of the formulation.

The primary objective of this study was to assess whether the micellar formulation of milk thistle could achieve superior bioavailability compared to the conventional formulation. To the best of our knowledge, this is the first human trial that utilizes a micellar silymarin formulation.

By determining the relative pharmacokinetic benefits of these formulations, we aim to provide critical insights into the potential for micellar systems to optimize silymarin-based therapies. In doing so, this research has the potential to pave the way for more effective treatments for liver disorders and other conditions where silymarin may offer clinical benefit.

In addition to pharmacokinetic analysis, this study also monitors the tolerability of the novel LipoMicel formulation. While milk thistle containing silymarin has a long history of use and has shown no major toxicity in animal studies, evaluation of safety and toxicity in new and more bioavailable formulations is still crucial [37,38,39].

This research highlights the importance of advanced formulation technologies in improving the therapeutic efficacy of natural compounds.

## 2. Materials and Methods

### 2.1. Study Design

This study employed a double-blind, randomized, crossover design to compare the bioavailability of two milk thistle formulations: an unformulated milk thistle extract (standard treatment (STD)) and a novel micellar, standardized, lipid-mediated milk thistle formulation (LipoMicel Milk Thistle (LMM)). It was managed and conducted in the late spring to winter months (May to January) at the research facility of ISURA (Burnaby, BC, Canada) under the supervision of the investigators. The study adhered to the CONSORT 2010 guidelines for reporting clinical trials. All subjects provided written informed consent before participation. The protocol was approved by the Canadian Shield Ethics Review Board (REB 2022-11-002) with OHRP Registration IORG0003491, FDA Registration IRB00004157. The study was registered at ClinicalTrials.gov [trial registry number: NCT06882681].

### 2.2. Participants

Twenty adult volunteers were initially screened, and 16 volunteers were enrolled and randomized to the treatments (Figure 2). Participants were excluded if they had any chronic diseases, were pregnant or lactating, had a history of gastrointestinal disorders, or were taking medications known to affect liver function or interfere with milk thistle metabolism. Inclusion and exclusion criteria were assessed during a screening visit. Participants were instructed to refrain from consuming alcohol, nicotine and/or other substances, and herbal supplements for 12 h prior to each dose administration, as well as during the study period.

Written informed consent was obtained from all participants prior to enrollment. Forty-eight hours before each treatment and during the treatment period, participants were instructed to refrain from taking supplements containing milk thistle and to maintain a normal, balanced diet.

The inclusion criteria included a signed written informed consent form and a willingness to avoid the consumption of any herbal supplements that contain milk thistle during the study period. Volunteers with serious acute or chronic diseases—such as liver, kidney, or gastrointestinal diseases—which may affect absorption, metabolism, and/or elimination of the treatment, as well as any kind of contraindication and/or allergy to milk thistle were excluded. Female participants must not have been pregnant, be planning pregnancy, or be breast-feeding. Participants had to complete an online health questionnaire on their medical history upon study enrollment.

### 2.3. Intervention

On treatment days, participants received a single oral dose of each milk thistle formulation (STD or LMM) according to each person’s allocated sequence along with a glass of water (approx. 200 mL) and a standardized breakfast. There was a washout period of 1 week between the two treatments. The two formulations were both administered in capsule form (Table 1). While participants were aware that two different formulations with potentially different bioavailability were being tested, they were blinded to the identity and composition of the products. However, the standard treatment (STD) was administered in a hard-gel capsule while the other treatment (LMM) was in a soft-gel capsule.

Unformulated/standard silymarin (STD: Milk Thistle, Lot 100284 from Organika Health Products, Richmond, BC, Canada) was purchased from Amazon.ca. One hard-gelatin capsule contained 130 mg of silymarin extracted from seeds of *Silybum marianum*.The new micellar delivery system of silymarin (LMM: LipoMicel^®^ Milk Thistle, Lot 2001076) was provided by Natural Factors, Coquitlam, BC, Canada. One soft-gel capsule contained 130 mg of silymarin extracted from seeds of *Silybum marianum* along with medium-chain triglycerides and food-grade components of the micellular matrix.

The LMM formulation (LipoMicel^®^) is an anhydrous, self-assembling delivery system that forms micelles in situ upon exposure to gastrointestinal fluids. As such, micelle formation does not occur prior to ingestion, and the formulation exists as a homogeneous liquid mixture of the active ingredient and food-grade excipients, including medium-chain triglycerides and phospholipids. Because the micelles are not pre-formed, conventional parameters such as micelle size, zeta potential, or encapsulation efficiency are not applicable in the pre-dosing state. However, the formulation was manufactured under Good Manufacturing Practices and tested according to Health Canada’s Quality of Natural Health Products Guide, including assessments of active ingredient content, identity, and purity.

With bioavailability of the complete formulation, including the combined effects of both the silymarin and the excipients, being the primary focus of this study, an excipient-only control group was not included. While MCT and phosphatidylcholine in the LMM formulation are known to influence absorption, these excipients are part of the micellar matrix to be studied as a whole and not separately.

Previous human studies evaluating silymarin bioavailability used dosages that ranged from 120 to 140 mg of silymarin [21,40]. Thus, a dosage of 130 mg silymarin was chosen for this study. This dosage is also typical of single-serving doses in commercially available milk thistle supplements [41].

### 2.4. Randomization and Blinding

Participants were randomly assigned to receive the test (micellar silymarin) and control (standard silymarin) formulations in a 1:1 allocation ratio using a crossover design. The randomization sequence was generated using Microsoft Excel’s random number function (Rand()). Each participant was assigned a unique random number, which was then ranked to determine treatment order. The finalized sequence was securely stored by an independent study assistant who was not involved in recruitment, data collection, or outcome assessment, thereby ensuring allocation concealment until after data analysis and minimizing the risk of selection bias.

The study participants, study coordinator, and principal investigator were blinded to the treatments throughout the study. Blinding was maintained by packaging both formulations in identical opaque bottles labeled with a study-specific participant identifier.

### 2.5. Safety and Tolerability

Potential adverse events associated with the study treatments were evaluated using a structured questionnaire administered during the 24 h post-dose monitoring period. The form included a checklist of predefined symptoms commonly associated with oral supplements (e.g., bloating, diarrhea, constipation, heartburn, nausea, etc.), each rated for severity on a 5-point scale (no symptoms, mild, moderate, severe or life-threatening) (Appendix A). Participants were also asked about the duration of any symptoms, whether daily activities were affected, whether additional medication was required, and if they had informed their healthcare provider. An open-ended field allowed participants to report any other symptoms not listed.

Given that the study involved a single-dose administration of a compound with a well-established safety profile, this self-report system was used as the primary method for adverse event monitoring. In addition, several well-established safety markers, including liver and kidney function parameters, were measured at baseline (0 h) and post-dose using capillary blood samples and analyzed with an SD-1 Auto Dry Biochemistry Analyzer (Seamaty Technology Co., Ltd., Chengdu, China). All blood samples were analyzed immediately after collection to maintain sample integrity and minimize variability. The Seamaty SD-1 Analyzer was calibrated according to the manufacturer’s standards to ensure the reliability of results.

### 2.6. Blood Sampling and Analysis

Blood samples were collected from participants at baseline (0 h) and at the following time points: 0.5, 1, 2, 3, 4, 6, 8, 10, 12, and 24 h post-dose. Capillary whole blood samples were drawn into K_3_ EDTA tubes (Sarstedt, Nümbrecht, Germany) and stored at −20 °C until analysis.

Whole blood concentrations of silymarin (including silybin A and B, isosilybin A and B, and other related compounds/metabolites) were determined using liquid chromatography-high-resolution mass spectrometry (LC-HRMS) on a Thermo Scientific Q-Exactive Orbitrap instrument. Samples were digested by enzymatic hydrolysis with β-glucuronidase to quantify total silymarin including free and conjugated forms and were then extracted with Acetonitrile and sonicated for 1 h. Chromatographic separation was performed on an Agilent Poroshell 120 EC-C18 column. Liquid chromatography was carried out with a binary solvent gradient progressing from 25% B to 90% B in 3.6 min and equilibrated for 4.5 min before the next injection. The mobile phases were 0.5% formic acid in water in A and 0.5% formic acid in Acetonitrile in B. The separation was performed at a flow rate of 400 µL/min.

The orbitrap mass spectrometer was operated with negative heated electrospray ionization in Full MS mode with a resolution setting of 70,000, a scan range of 200–1500 *m*/*z*, and a maximum ion trapping time of 100 milliseconds. The typical drift in mass accuracy over a one-week period is less than 1 ppm, and chromatographic data was extracted with a mass tolerance of 5 ppm. Circulating Silybin, Silychristin, Isosilybin, Isosilychristin, and Silydianin were detected as their deprotonated molecular ion (C_25_H_21_O_10_^−^, *m*/*z* = 481.1140).

The quantification method was validated for accuracy, precision, and selectivity, with the calibration curve showing good linearity (R^2^ > 0.995) over the concentration range of silymarin. The limits of detection (LOD) and quantification (LOQ) were 0.34 ng/mL and 1.1 ng/mL, respectively.

Data were collected using Xcalibur^TM^ 5.0 (Thermo Fisher Scientific Inc., Waltham, MA, USA) and analyzed with TraceFinder 5.0 (Thermo Fisher Scientific Inc., Waltham, MA, USA) software with the default mass tolerance set to 5.00 ppm.

Concentrations of silymarin in capillary whole blood were determined based on internal standard calibration with a 6-point calibration curve using Silybin (USP, Rockville, MD, USA) as the chemical reference standard and epicatechin as the internal standard (Certified Reference Material, secondary standard, Millipore Sigma, Burlington, MA, USA).

### 2.7. Pharmacokinetic Analysis

Pharmacokinetic parameters were determined using non-compartmental analysis (NCA) via the software PKSolver (version 2.0) [42]. The following pharmacokinetic parameters were calculated for both formulations: maximum plasma concentration (C_max_), time to reach maximum concentration (T_max_), area under the plasma concentration-time curve (AUC) from 0 to 24 h (AUC_0–24_), elimination half-life (T_1/2_), and mean residence time (MRT).

### 2.8. Outcome Measures

The primary outcome measures were the area under the plasma concentration-time curve (AUC) from 0 to 24 h (AUC_0–24_), the maximum plasma concentration (C_max_), the time to reach maximum concentration (T_max_), as well as the elimination half-life (T_1/2_) and the mean residence time (MRT) of silymarin for each formulation.

Additionally, safety and tolerability were monitored by means of a health questionnaire that asked the participants to report any symptoms of bloating, constipation, diarrhea, heartburn, abdominal pain or cramps, rash, nausea, dizziness, or blurred vision.

### 2.9. Statistical Analysis

Pharmacokinetic (PK) parameters were analyzed with the Shapiro–Wilk test for normal distribution and for lognormal distribution. All PK parameters except for T _max_ passed the lognormal distribution test and were statistically analyzed using paired *t*-tests with Holm–Šídák correction for multiple comparisons after applying a log-transformation of the data; T_max_ values failed both normal distribution and lognormal distribution tests and were thus compared using the Wilcoxon matched-pairs signed rank test with Holm–Šídák correction for multiple comparisons. The Holm–Šídák correction for multiple comparisons was applied to primary PK outcomes (C_max_, AUC_0–24_, T_max_, MRT, and T_1/2_), and both adjusted and unadjusted *p*-values are reported in Appendix A. The Holm–Šídák method was selected over more conservative approaches such as Bonferroni because it provides uniformly greater statistical power while still controlling the family-wise error rate. This stepwise, sequentially rejective procedure accounts for the number and ordering of comparisons, making it particularly appropriate for studies with a small set of pre-specified, potentially correlated pharmacokinetic outcomes. Results are primarily presented as Geometric means (GM) with 95% confidence intervals (CIs); and arithmetic means (AM) ± standard deviation (SD) values are also reported for descriptive purposes. A *p*-value < 0.05 was considered statistically significant. All statistical analyses were performed using GraphPad Prism (version 10.4.0, Dotmatics, Boston, MA, USA).

Given the complexity of the formulation, we interpret the enhanced bioavailability of LMM as likely reflecting the combined effect of both the silymarin and the micellar delivery system. We believe the observed differences in bioavailability provide important insights into the potential for lipid-based micellar systems to improve the oral absorption of poorly soluble compounds like silymarin.

### 2.10. Sample Size and Power

Sample size was determined based on a power calculation using C_max_ data from a recent human study comparing two silymarin formulations [43]. Based on a calculated effect size of 0.706, a total of 13 participants were required to achieve 80% power at a significance level of α = 0.05 for a matched-pairs *t*-test. Sample size estimation was conducted using G*Power (version 3.1.9.7, Heinrich Heine University Düsseldorf, Germany).

### 2.11. Cryo-SEM Sample Preparation and Imaging

Around 400 mg LMM soft-gel fill material was dispersed in deionized water to make 1.5 mL suspension in a 1.5 mL polypropylene microcentrifuge tube with snap cap. The suspension was sonicated in a warm bath (30–40 °C) for 15 min and then allowed to settle for 5 min. A few drops of light-colored top portion of the suspension were filled into wells made on an aluminum cryo-SEM holder with a small amount of overfill. The cryo-SEM holder with the sample was then submerged in a slushy liquid nitrogen for 10–20 s to rapidly freeze the samples. After freezing, the sample was vacuum transferred into a Quorum PP3010T cryochamber (Quorum Technologies, East Sussex, UK) to fracture the overfill portion off in order to reveal the cross-section of the frozen sample. The fractured sample was then further transferred into a Helios NanoLab 650 scanning electron microscope (FEI Company, Hillsboro, OR, USA) for imaging. Cryo-SEM images were collected with a current of 13 pA at 2 kV, with a working distance of 4 mm, at a scanning resolution of 3072 × 2207 or lower by averaging 128 low-dose scanning frames with drift correction. The sample was kept at −140 °C when fracturing and imaging. The sample was also imaged after sublimation at −80 °C for 15 min in a cryo-SEM chamber to remove some water.

### 2.12. Particle-Size Distribution

To characterize the self-assembled micelles formed by the LMM formulation under aqueous conditions, particle-size distribution was assessed using laser diffraction. Each formulation was dispersed in water to simulate gastrointestinal fluid exposure, and particle sizes were measured using a Mastersizer 3000 particle size analyzer (Malvern Panalytical, Quebec City, QC, Canada). For the STD powder, the hydrodynamic size distribution reflects the particles formed upon aqueous dispersion, consistent with common pharmaceutical practice in evaluating dissolution behavior and bioavailability. Approximately 1 mL of each formulation was added to the Hydro SM wet dispersion unit containing 200 mL of water under continuous stirring. Measurements were taken after the dispersion reached an obscuration of ~10%, and data were collected over a 1 min period with continuous circulation through the optical cell.

Hydrodynamic diameters were determined from diffraction data using Mastersizer software (v3.81). To describe the size uniformity of the micelles, the size span was calculated using the formula:$$Size\;Span=\frac{D_{90\%}-D_{10\%}}{D_{50\%}}$$
where D_N%_ reflects the percentage of particles (N%) with a diameter (D) less than or equal to the specified value. A monodisperse (uniform) micellar population has a low value for its size span, whereas higher values suggested heterogeneity in particle size [44].

## 3. Results

### 3.1. Participant Demographics

Twenty volunteers were initially screened from May to July of 2024 according to the inclusion and exclusion criteria. A total of 16 participants (8 males, 8 females) were enrolled in the study. Since all participants completed both treatment phases, and no participants were lost to follow-up, the baseline characteristics for participants in both interventions are identical (See Figure 2 for the flow diagram).

The mean age of the participants was 39.5 ± 2.7, with a mean body mass index (BMI) of 23.7 ± 0.6 kg/m^2^ (Table 2).

### 3.2. Pharmacokinetic Outcomes

The pharmacokinetic parameters for the two treatments are summarized in Table 3. Geometric means (GM) with 95% confidence intervals (CIs) are provided as the primary results, as they are derived from the log-transformed analysis. Additionally, arithmetic means (AM) are included for reference and comparison with previous studies. Statistical comparisons were conducted on log-transformed values, with the results transformed back to the original scale for interpretability.

The log-transformed analysis showed that C_max_ and AUC_0–24_ were significantly higher in the LMM treatment group compared to STD (Figure 3 and Figure 4). Specifically, the AUC_0–24_ was approx. 11-fold higher in the LMM group, with a statistically significant difference between formulations (*p* = 0.015). Similarly, C_max_ values were approx. 19-fold higher with the LMM formulation, reaching peak concentration five times faster than the standard formulation (T_max_: 0.5 vs. 2.5 h; *p* = 0.015). The arithmetic means, while higher, exhibited greater variability due to the influence of extreme values.

Additionally, the standard formulation exhibited a significantly longer (up to approx. two times) mean residence time (MRT) compared to the LMM formulation (*p* = 0.015), suggesting differences in absorption and clearance dynamics. However, the T_½_ levels were not significantly different between the standard formulation and LMM.

### 3.3. Side Effects

A safety study involving 13 participants (7 males and 6 females) found that both formulations were well tolerated by all participants. Participants reported no adverse events throughout both interventions. While there were some statistically significant changes in total bilirubin (with STD and LMM treatment) and triglyceride (with LMM treatment) after 24 h, the changes remained within the normal range (Table 4 and Table 5). LDL and TC were above normal after STD treatment but the baseline for both these markers was already elevated.

### 3.4. Cryo-SEM

Cryo-SEM analysis revealed that the micelles of LMM are round and range in size from 10 to 80 µm in diameter, which contrasts sharply with the smooth, flat surfaces observed in the unformulated/standard extract (STD) (Figure 5). This distinct difference in morphology suggests that the LMM formulation facilitates micelle formation, which may contribute to its enhanced solubility and bioavailability. The spherical micellar structures observed in LMM are consistent with self-assembling lipid-based delivery systems, which can improve intestinal absorption by facilitating dispersion and mucoadhesion. In contrast, the smooth, dense morphology of STD may suggest poor aqueous dispersion, which is likely a contributing factor to its lower absorption.

### 3.5. Particle-Size

Upon aqueous dispersion, LMM produced particles with a median hydrodynamic diameter of approximately 255 µm and a size span of 0.335, indicating a relatively uniform particle population. While the measured size exceeds the nanoscale range typical of classical micelles, these results are consistent with the formation of stable colloidal structures upon dispersion, supporting the self-assembling nature of the delivery system under simulated physiological conditions. On the other hand, STD produced particles with lower median hydrodynamic diameter of 98.1 µm but a larger size span of 2.90 with a polydisperse profile (Figure 6 and Table 6). See Discussion for clarification regarding the terminology and interpretation of these values.

## 4. Discussion

Despite promising pharmacology, silymarin’s poor oral bioavailability limits its clinical use. This study evaluated a novel self-assembling micellar system (LipoMicel, LMM) designed to enhance solubility and absorption, compared to standard milk thistle extract (STD). While not designed to assess long-term safety, no adverse events or clinically significant changes in blood chemistry were observed during the 24 h study period (Table 4 and Table 5).

Although the term micellar formulation is used throughout this study to describe LipoMicel^®^, the observed particle sizes (~255 μm by laser diffraction; 10–80 μm by cryo-SEM) far exceed classical micelle dimensions (10–100 nm). This likely reflects the aggregation of smaller micellar subunits into larger colloidal or vesicular structures—common in polydisperse, amphiphilic systems. Size readings may also be skewed by laser diffraction’s bias toward larger particles and potential cryo-SEM artifacts. While not a strict structural term here, “micelle” reflects the product’s amphiphilic self-assembly and aligns with the manufacturer’s and prior literature’s usage. Nevertheless, the pharmacokinetic results showed significant differences, providing more in-depth insights into their absorption, distribution, and elimination characteristics. Specifically, the LMM formulation demonstrated faster absorption (T_max_), higher peak blood concentrations (C_max_), as well as greater total exposure (AUC_0–24_), while the standard formulation had a longer mean residence time (MRT) (Table 3).

The observed differences in mean residence time (MRT) between the two formulations suggest distinct absorption and clearance profiles. Interestingly, while the STD formulation exhibited a significantly longer MRT (approximately two times), the T_1/2_ concentrations were not significantly different from LMM. This longer MRT likely reflects slower absorption and delayed systemic clearance, possibly due to lower solubility and slower gastric absorption rates, and it supports the hypothesis that the STD formulation results in slower systemic clearance, likely due to prolonged absorption and metabolic processes.

Interestingly, the LMM formulation exhibited a significantly higher C_max_, achieved five times faster (significantly lower T_max_, *p* = 0.015). This suggests that the LMM formulation enhances the absorption of silymarin, resulting in faster peak systemic uptake and a shorter MRT. This rapid absorption and more efficient clearance from the system suggests that the micellar delivery system facilitates a faster transit and utilization of silymarin.

Additionally, the observed increase in bioavailability (up to 11.4-fold higher AUC_0–24_) suggests that the micellar formulation improves silymarin absorption and intestinal uptake. This may also lead to more efficient hepatic metabolism and systemic clearance, as the micelles allow better solubilization and absorption of silymarin in the gastrointestinal tract. Despite the similar T_1/2_ values between LMM and STD, the increased AUC and C_max_ for LMM point to an overall more efficient bioavailability profile, which could be beneficial in clinical applications requiring more immediate therapeutic effects. This improved bioavailability does not appear to result from a reduction in hydrodynamic particle size since LMM was observed to have a median hydrodynamic particle diameter that is more than two times that of STD (255 µm vs. 98.1 µm, Table 6).

Supporting this, cryo-SEM imaging revealed the formation of colloidal aggregates and self-assembled structures upon aqueous dispersion of the LMM formulation. These morphologies are consistent with lipid-based carriers known to enhance solubility and absorption and likely contributed to the observed improvements in systemic exposure.

On the other hand, the STD formulation, despite its lower C_max_ and AUC_0–24_, maintains a longer systemic retention of silymarin due to its prolonged MRT. This suggests that the STD formulation may be more suitable for chronic conditions where sustained exposure to silymarin is beneficial. The slower absorption and longer duration of action could be advantageous in therapeutic settings where continuous, gradual release of the active compound is desired.

Although T_1/2_ values were not significantly different, this does not rule out faster systemic clearance with the LMM formulation. T_1/2_ reflects both absorption and elimination, and faster uptake does not always translate to a shorter half-life—especially with complex hepatic and renal clearance. Future studies should explore these pathways to better understand how micellar delivery affects silymarin’s systemic elimination.

To summarize, while the STD formulation maintains prolonged systemic exposure, it does so at the cost of lower bioavailability. In contrast, the LMM formulation provides faster absorption, higher bioavailability, and more rapid clearance, making it a potentially more effective delivery system in acute clinical situations that require a quicker onset of silymarin activity. However, further studies exploring the clearance mechanisms of LMM will be necessary to fully understand the implications for systemic clearance and the overall therapeutic potential of the formulation.

Our findings align with and expand upon previous studies examining the bioavailability of milk thistle formulations. For example, a review by Costanzo et al. cited multiple examples of lipid-based formulations of silymarin including liquid oil-in-water emulsions and solid-lipid particles that showed improved bioavailability compared to standard unformulated extracts, which is consistent with our observation of a higher C_max_ and AUC for the LMM formulation [14]. Similarly, several animal studies demonstrated that lipid-mediated formulations of milk thistle are absorbed more rapidly and exhibited greater efficacy [13,17,18,19].

Previous human pharmacokinetic studies using similar doses of silymarin also reported comparable C_max_ ranges as in the current study. For example, Barzaghi et al. reported a C_max_ value of 102 ± 22 ng/mL of silybin after an oral dose of 360 mg of unformulated silymarin, while Zhu et al. reported a C_max_ value of 106.9 ± 49.2 ng/mL after an oral dose of 140 mg [40,45]. For comparison, our study found a mean C_max_ value of 66.1 ng/mL (95% CI: 4.9–127.3) for the 130 mg unformulated silymarin, albeit with a high inter-subject variability. Compared to other formulated silymarin products, a phospholipid complex of silymarin (phytosome) was reported to have 4.6 times higher bioavailability compared to unformulated silymarin; furthermore, a self-microemulsifying drug delivery system (SMEDDS) was reported to provide 1.7–2.5 times higher silymarin AUC than unformulated silymarin [21,40,46]. Based on AUC_0–24_, LMM in our study was found to be approx. 11 times more bioavailable compared to unformulated silymarin, which may indicate superior bioavailability to the other formulations. However, further studies of the various formulated products on the same group of participants would be recommended to support a direct comparison of bioavailability between different formulations.

In terms of rates of absorption, conventional silymarin formulations typically reach T_max_ at about 2–3 h, which is within the range of the 2.5 h (90% CI: 0.5–8.0) we reported for STD in this study (Table 3) [2,47]. While a phytosome formulation reported delayed T_max_ (from 4 to 6 h) in humans, and formulations with piperine and fulvic acid delayed T_max_ from 1.0 h to 1.5 h in rats, the LMM formulation in the current study did not exhibit a delayed T_max_. On the contrary, LMM showed reduced T_max_ in a manner similar to SMEDDS formulations [15,17,21,47,48].

### Clinical Relevance and Potential Applications

This study highlights the potential of the LipoMicel^®^ (LMM) micellar formulation to enhance silymarin bioavailability, with a significantly higher C_max_ and faster T_max_ compared to a standard extract. These characteristics suggest LMM may be particularly useful in situations requiring rapid therapeutic actions—such as acute liver stress or inflammation—by achieving quicker and stronger effects at lower doses. However, its shorter mean residence time may necessitate more frequent dosing to maintain therapeutic levels. While no adverse events or blood chemistry changes were observed in this 24 h study, the higher peak levels raise considerations for safety and dose optimization in broader clinical use. Moreover, the formulation’s improved absorption may benefit individuals with impaired bile production or digestion, who may not absorb unformulated silymarin efficiently.

Although labeled as “micellar,” the LMM formulation forms larger colloidal aggregates (>100 μm), likely due to the self-assembly of amphiphilic excipients in aqueous media. This is consistent with lipid-based systems that require physiological conditions for full micellization. The enhanced pharmacokinetics may thus stem from both micelle formation and the excipients’ independent bioenhancing effects (e.g., MCTs, phosphatidylcholine), though the lack of excipient-only control limits mechanistic clarity. Additionally, differences in capsule appearance may have compromised blinding, and the study did not include comparisons with other advanced silymarin formulations like phytosomes. While these factors do not alter the core findings, they limit broader conclusions about superiority or mechanism. Future studies should address these gaps using physiologically relevant media, matched placebo controls, and broader formulation comparisons to validate and contextualize these promising results.

The pharmacokinetic differences observed in this study suggest that the LMM formulation may be better suited for rapid onset of action, such as in scenarios requiring acute liver support or a quick therapeutic response. The higher C_max_ and faster T_max_ observed with the LMM formulation imply that it may deliver more immediate effects, and stronger effects at lower doses which could be advantageous for acute treatments. However, the shorter mean residence time might necessitate more frequent dosing to maintain optimal therapeutic levels over time.

Although the LMM formulation is designed to self-assemble into micelles following oral ingestion, the particle-size measurements obtained under aqueous dispersion conditions showed a mean hydrodynamic diameter of approximately 115 µm. This value exceeds the typical size range of classical micelles (10–100 nm) and likely reflects the formation of larger colloidal aggregates or emulsion-like structures upon dispersion in water. Such behavior is not uncommon in anhydrous, lipid-based systems that require physiological conditions (e.g., bile salts, digestive enzymes) for complete micellization. Therefore, while the current in vitro measurements support the formation of stable, self-assembled particles, further work using biorelevant media or dynamic gastrointestinal models would be required to fully characterize the micelle formation process under physiological conditions. These discrepancies between in vitro and in vivo conditions may limit mechanistic interpretations regarding how micelle formation contributes to absorption. Therefore, the enhanced bioavailability observed in vivo should be interpreted with caution, as it may involve dynamic transformations not fully captured in static in vitro assays.

Also, the differences in pharmacokinetic profiles observed between the STD and LMM formulations may be partly attributed to differences in excipient composition. The LMM formulation incorporates amphiphilic excipients designed to form colloidal assemblies that enhance solubility, membrane permeability, and lymphatic transport of silymarin flavonolignans. In contrast, the STD formulation consists of unformulated silymarin powder, which lacks solubilizing agents and is thus more reliant on endogenous bile salts for dissolution. These compositional differences likely contributed to the improved systemic exposure and earlier T_max_ observed with LMM. However, the absence of an excipient-only control arm limits our ability to distinguish whether these effects are attributable primarily to micelle formation or to independent bio-enhancing properties of the excipients themselves. This may have led to an overestimation of the specific contribution of micellization to the observed pharmacokinetic improvements. Similar excipient-mediated effects have been reported in other studies employing micellar or lipid-based delivery systems for poorly water-soluble phytochemicals.

While the LipoMicel formulation demonstrated markedly enhanced bioavailability compared to the unformulated one, this comparison does not include other advanced formulations such as phytosomes. The lack of direct comparison with other advanced formulations limits the generalizability of our findings and precludes definitive conclusions about the relative superiority of LMM among available silymarin delivery platforms. Therefore, although the results underscore LMM’s potential to improve silymarin absorption, further studies are warranted to evaluate its performance relative to other established delivery technologies. Additionally, due to the anhydrous, self-assembling nature of the micellar formulation, it was not characterized using standard colloidal parameters such as zeta potential or encapsulation efficiency. Future studies could incorporate in vitro dissolution or dynamic micelle formation studies in simulated gastrointestinal fluids to provide further mechanistic insight into the absorption process.

One limitation of this study is the potential compromise in blinding due to differences in the appearance of the treatment formulations—specifically, the soft-gel (LMM) vs. hard-gel (STD) capsules. While participants were not informed of which treatment they received, the visible differences between the two formulations may have introduced unintentional bias. Given that pharmacokinetic parameters are objective measures unlikely to be influenced by unblinding, this limitation is unlikely to have affected the primary outcomes. Nevertheless, it remains a potential source of participant or observer bias for any subjective endpoints, should these be evaluated in future trials. More robust blinding strategies, such as using capsules of the same appearance, could help to mitigate this risk. Furthermore, future studies with larger samples and broader populations may further validate these results.

Additionally, excipients such as MCTs and phosphatidylcholine may independently enhance the bioavailability of lipophilic compounds, but an excipient-only control group was not included. Future studies could isolate the excipient effects by testing an excipient-only group (with MCTs and phosphatidylcholine, but no silymarin) to further clarify the independent contribution of these components.

Although the LMM formulation achieved a higher peak concentration (C_max_), the faster elimination of the compound may mitigate potential risks of accumulating excessive levels of silymarin. While higher C_max_ may be beneficial for rapid therapeutic action, it also raises safety considerations, especially in sensitive populations or when used concomitantly with other hepatically metabolized drugs. Thus, dosing optimization studies will be essential.

In summary, while these limitations do not undermine the core pharmacokinetic findings, they may influence mechanistic interpretations, safety extrapolations, and comparative generalizations. Future studies incorporating physiological simulation, control groups, matched blinding, and broader formulation comparisons will be essential to confirm and contextualize these results.

These pharmacokinetic enhancements have practical clinical implications. In scenarios such as acute liver injury, intoxication, or flare-ups of chronic liver disease, a formulation like LMM that achieves higher plasma levels more rapidly could lead to more immediate therapeutic action. Furthermore, if higher bioavailability can be sustained at lower doses, LMM may reduce pill burden and improve adherence, particularly in populations with polypharmacy concerns, such as elderly patients or those with chronic comorbidities. Additionally, improved absorption may enhance clinical consistency in populations with compromised bile production or impaired digestion, who may absorb unformulated silymarin less efficiently. Finally, given silymarin’s reported anti-inflammatory, antioxidant, and hepatoprotective properties, enhanced systemic exposure might broaden its utility in adjunctive roles beyond hepatic support, such as metabolic syndrome or chemotherapy-induced toxicity, though this remains to be tested.

## 5. Conclusions

In summary, the LipoMicel milk thistle formulation (LMM) exhibited significantly faster absorption, higher peak plasma concentrations, and greater overall exposure (AUC_0–24_) compared to the unformulated product, but it has a shorter mean residence time. These differences suggest that the LMM formulation may be better suited for scenarios requiring rapid onset of action, while the standard formulation may be more appropriate for sustained therapeutic effects. Further studies evaluating the clinical outcomes of these pharmacokinetic differences are needed to determine the optimal formulation for specific therapeutic indications.

## Figures and Tables

**Figure 1 pharmaceutics-17-00880-f001:**
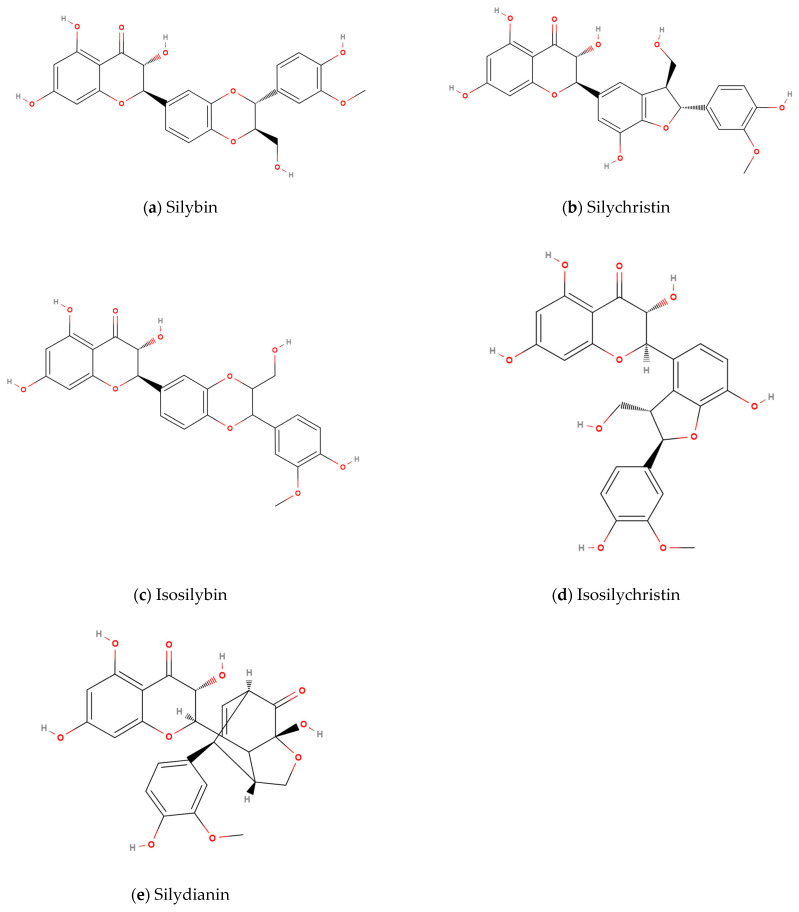
Chemical structures of silymarin flavonoids. (**a**) Silybin, (**b**) Silychristin, (**c**) Isosilybin, (**d**) Isosilychristin, and (**e**) Silydianin. These compounds share the same molecular formula (C_25_H_22_O_10_), have a common flavonolignan backbone, but differ in hydroxylation and stereochemistry, which may influence their solubility, absorption, and biological activity. Silybin (**a**) is the most abundant and pharmacologically active component, making it a key marker for evaluating silymarin bioavailability in this study.

**Figure 2 pharmaceutics-17-00880-f002:**
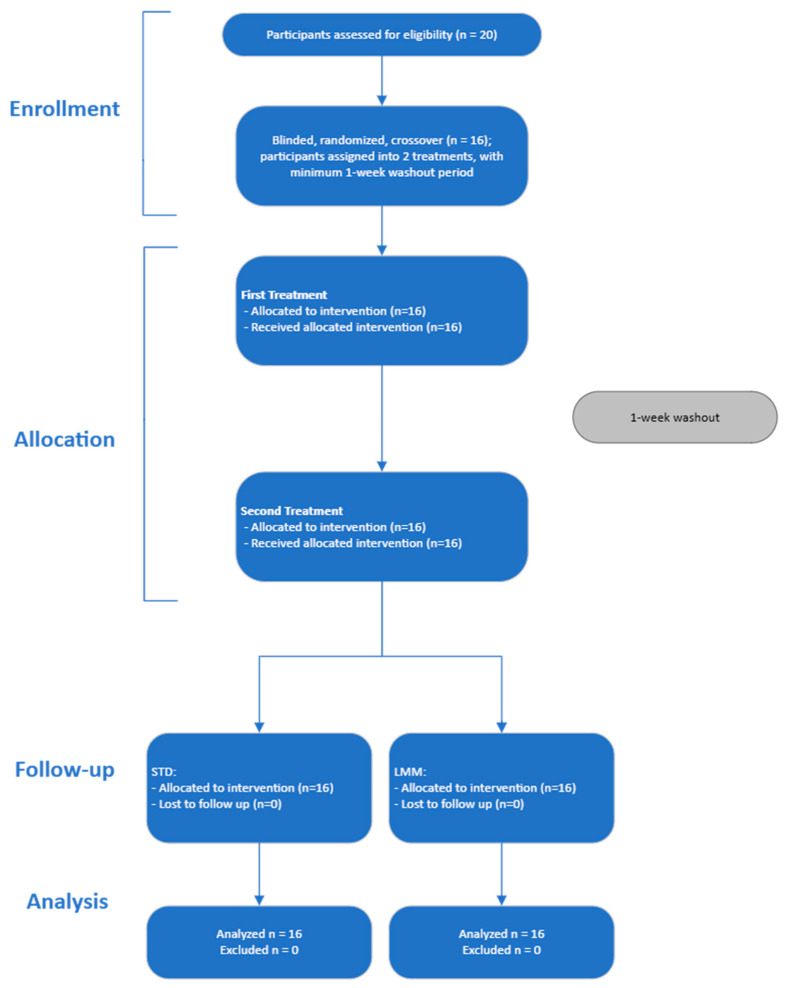
Study flow diagram of the randomized, double-blind, crossover clinical trial evaluating the bioavailability of two silymarin formulations. A total of 20 participants were screened, and 16 eligible participants were enrolled and randomized to receive either the standard (STD) or micellar (LMM) formulation in a crossover design with a 1-week washout period. All participants completed both treatment phases and were included in the pharmacokinetic and safety analyses.

**Figure 3 pharmaceutics-17-00880-f003:**
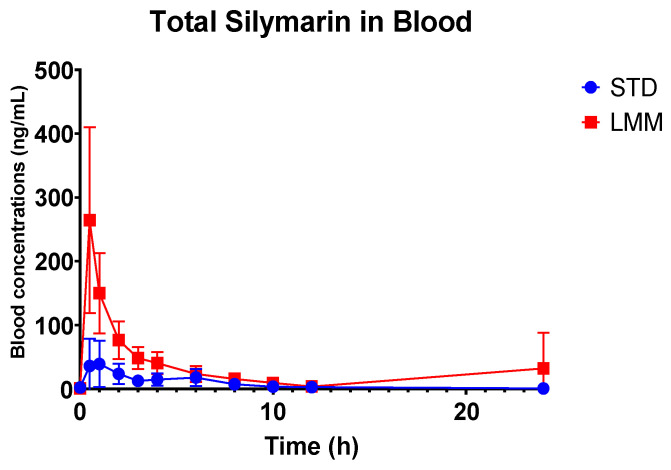
Mean whole blood concentrations of total silymarin (free + conjugated) over 24 h following a single oral dose of either the standard formulation (STD; blue) or the micellar formulation (LMM; red). The LMM group exhibited significantly higher systemic exposure at all time points. Data are presented as mean ± 95% confidence intervals (*n* = 16). Statistical analysis showed significant differences between treatments (*p* = 0.006), across time points (*p* = 0.001), and for the treatment × time interaction (*p* < 0.0001).

**Figure 4 pharmaceutics-17-00880-f004:**
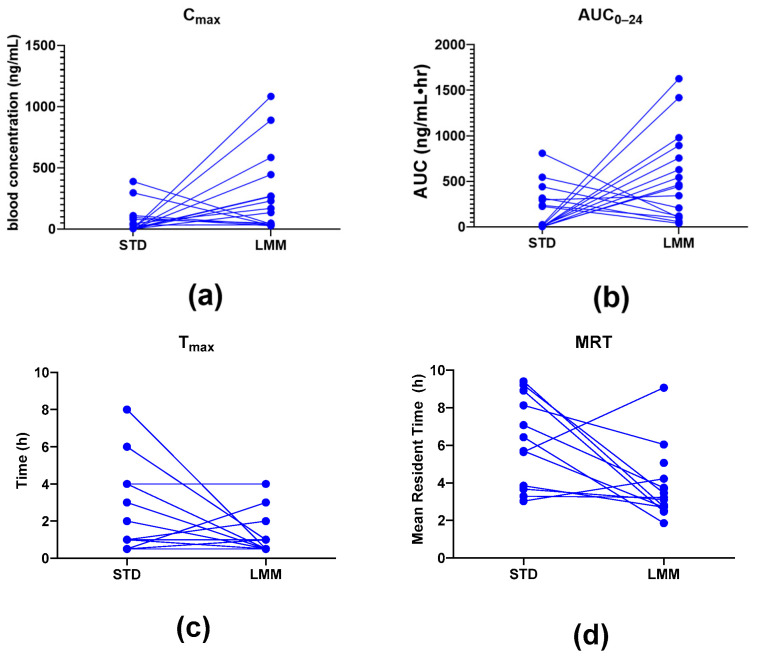
Individual participant pharmacokinetic responses (*n* = 16) comparing the standard (STD) and micellar (LMM) silymarin formulations. Each panel displays paired data for: (**a**) maximum plasma concentration (C*_max_*), (**b**) total systemic exposure (AUC_0–24_), (**c**) time to reach peak concentration (T*_max_*), and (**d**) mean residence time (MRT). While most participants exhibited higher C*_max_* and AUC_0–24_ and faster T*_max_* with LMM, inter-individual variability was observed. These trends support enhanced absorption with the LMM formulation in the majority of participants.

**Figure 5 pharmaceutics-17-00880-f005:**
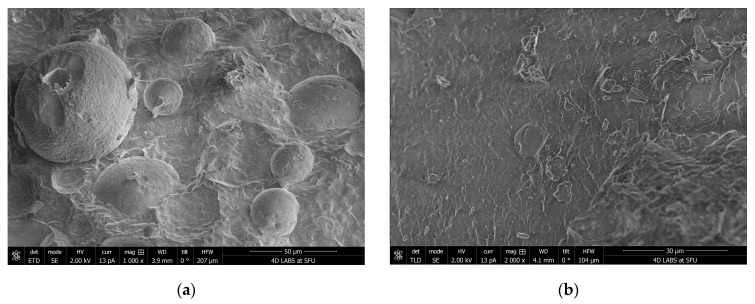
Cryo-scanning electron microscopy (Cryo-SEM) images of silymarin formulations after aqueous dispersion. (**a**) LipoMicel Milk Thistle (LMM) shows spherical, self-assembled colloidal structures consistent with micelle-like aggregates. (**b**) Standard Milk Thistle (STD) displays dense, irregular morphology with poor dispersion. These structural differences support the enhanced solubility and absorption observed with the LMM formulation in pharmacokinetic analysis.

**Figure 6 pharmaceutics-17-00880-f006:**
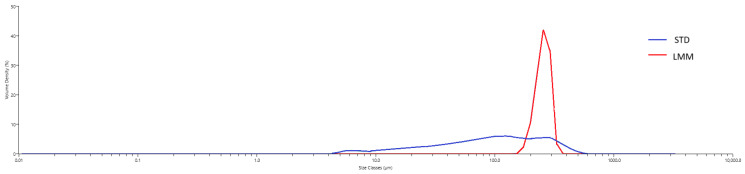
Hydrodynamic particle-size distribution of STD and LMM as determined through laser-diffraction following aqueous dispersion. LMM shows a monodisperse peak with larger average particle-sizes compared to STD. Measurements reflect the size of colloidal particles formed under simulated dispersion conditions, relevant to oral administration. For the STD formulation, this represents the particle size after suspension in water, not the raw dry powder. The LMM formulation exhibited a monodisperse profile with a larger median particle size (~255 µm), while the STD formulation showed a broader, polydisperse distribution with a smaller median size (~98 µm). These differences suggest improved colloidal stability and dispersion uniformity in the LMM formulation, which may contribute to its enhanced bioavailability.

**Table 1 pharmaceutics-17-00880-t001:** Study interventions.

Intervention	STD	LMM
Dosage form	Hard-gelatin capsules	Soft-gelatin capsules
Number of capsules per dose	1	1
Physical form of capsule content	Powder	Liquid
Total Silymarin per dose (mg)	130	130
Silychristin	34.3%	19.5%
Silydianin	1.7%	1.8%
Silybin A	15.9%	30.0%
Silybin B	28.9%	38.3%
Dehydrosilybin	2.3%	0.5%
Isosilybin A	11.8%	7.4%
Isosilybin B	5.1%	2.5%
Non-medicinal ingredients	Magnesium stearate (vegetable source), microcrystalline cellulose, brown rice flour, Hypromellose/Pullulan (vegetarian capsule)	Gelatin, glycerin, purified water, medium-chain triglycerides, MSM, xylitol, *Stevia rebaudiana* leaf extract, phosphatidylcholine lecithin.

STD: Standard Milk Thistle; LMM: LipoMicel Milk Thistle (novel micellar delivery system). Silybin A and B are diastereomeric pairs, as are Isosilybin A and B. In both formulations, the combined content of Silybin A and B represents the most abundant flavonolignan fraction, while Silydianin—the most structurally distinct component of silymarin—is present in the lowest proportion.

**Table 2 pharmaceutics-17-00880-t002:** Baseline characteristics of enrolled participants.

	Males	Females	Combined
N	8	8	16
Age	41.4 ± 3.1	37.5 ± 4.3	39.5 ± 2.7
Weight (kg)	74.5 ± 2.7	60.1 ± 2.8	67.3 ± 2.7
Height (cm)	173.6 ± 1.7	162.8 ± 2.5	168.2 ± 2.0
BMI (kg/m^2^)	24.7 ± 0.9	22.6 ± 0.6	23.7 ± 0.6

Values are presented as mean ± standard error of the mean (SEM). No significant differences were observed between groups at baseline.

**Table 3 pharmaceutics-17-00880-t003:** Pharmacokinetic parameters of total silymarin following administration of the standard and micellar formulations in 16 healthy participants ^1 2^.

	STD				LMM				
	AM	SD	GM	95% CI	AM	SD	GM	95% CI	*p*
T_1/2_ (h)	4.1	2.2	3.6	2.6–4.9	2.8	1.1	2.6	2.1–3.2	0.195
T_max_ (h)			2.0 ^1^	0.5–6.0 ^2^			0.5 ^1^	0.5–4.0 ^2^	0.035
C_max_ (ng/mL)	66.1	114.8	7.7	1.9–30.7	284.1	319.3	146.4	74.4–288.3	0.035
AUC_0–24_ (ng·h/mL)	183.6	244.9	28.9	7.4–113.5	544.6	485.5	330.4	178.3–612.5	0.046
MRT (h)	6.2	2.4	5.7	4.4–7.5	3.7	1.8	3.4	2.8–4.2	0.046

STD: Treatment with Unformulated Silymarin; LMM: Treatment with LipoMicel Silymarin; T_1/2_: elimination half-life; T_max_: time to reach C_max_; C_max_: maximum blood concentration of total silymarin observed during the study period; AUC_0–24_: the area under the blood concentration curve from the time of administration to 24 h; MRT: Mean Residence Time. Data are log-transformed and reported as Geometric Means (GM) with 95% confidence intervals (CIs); Arithmetic Means (AM) ± Standard Deviation (SD) are reported for descriptive purpose; T_max_ values are reported as Median (range)*. t*-test was used for paired pharmacokinetic parameters (log-transformed); T_max_ was analyzed by the Wilcoxon Signed-Rank test; Reported *p*-values were adjusted with Holm–Šídák multiple comparison correction. *p*-values < 0.05 were considered statistically significant. *n* = 16. Unadjusted *p*-values are reported in Appendix A. ^1^ Median value reported. ^2^ Range reported.

**Table 4 pharmaceutics-17-00880-t004:** Changes in clinical safety markers before and 24 h after administration of the standard silymarin formulation (STD).

	0 h	24 h	*p*-Value	Normal Range
Total Bilirubin (µmol/L)	17.4 ± 6.5	14.9 ± 7.0	0.0480	3.4–21.0
AST (U/L)	24.9 ± 10.2	24.5 ± 6.6	0.902	15.0–40.0
ALT (U/L)	28.3 ± 13.3	30.2 ± 16.3	0.384	9.0–50.0
Creatinine (µmol/L)	70.9 ± 9.7	69.6 ± 14.7	0.672	44.0–97.0
eGFR (mL/min/1.73 m^2^)	106.9 ± 11.3	109.4 ± 8.4	0.366	>90
HDL (mmol/L)	1.6 ± 0.3	1.5 ± 0.3	0.197	1.16–1.42
LDL (mmol/L)	3.4 ± 1.4	3.5 ± 1.3	0.125	0.50–3.14
TC (mmol/L)	5.5 ± 1.5	5.6 ± 1.4	0.584	0–5.17
TG (mmol/L)	1.3 ± 0.5	1.2 ± 0.5	0.553	0–1.70
GLU (mmol/L)	4.5 ± 0.6	4.6 ± 0.7	0.583	3.89–6.11
tCO2 (mmol/L)	25.9 ± 0.7	25.0 ± 2.0	0.121	22.0–29.0

Data are presented as mean ± standard deviation (SD). A paired *t*-test was used to assess statistical significance. All values remained within normal clinical reference ranges. *p* < 0.05 was considered statistically significant.

**Table 5 pharmaceutics-17-00880-t005:** Changes in clinical safety markers before and 24 h after administration of the micellar formulation (LMM).

	0 h	24 h	*p*-Value	Normal Range
Total Bilirubin (µmol/L)	11.7 ± 5.1	14.4 ± 6.3	0.0233	3.4–21.0
AST (U/L)	28.0 ± 14.3	26.7 ± 7.2	0.745	15.0–40.0
ALT (U/L)	29.8 ± 11.7	30.3 ± 14.5	0.873	9.0–50.0
Creatinine (µmol/L)	70.4 ± 9.8	73.8 ± 12.6	0.366	44.0–97.0
eGFR (mL/min/1.73 m^2^)	104.6 ± 14.1	102.8 ± 12.2	0.697	>90
HDL (mmol/L)	1.4 ± 0.5	1.6 ± 0.3	0.171	1.16–1.42
LDL (mmol/L)	3.6 ± 1.4	3.6 ± 1.3	0.989	0.50–3.14
TC (mmol/L)	5.7 ± 1.3	5.8 ± 1.4	0.126	0–5.17
TG (mmol/L)	1.5 ± 0.9	1.3 ± 0.7	0.042	0–1.70
GLU (mmol/L)	4.8 ± 0.9	4.8 ± 0.7	0.630	3.89–6.11
tCO2 (mmol/L)	25.0 ± 4.0	27.7 ± 4.5	0.129	22.0–29.0

Data are presented as mean ± standard deviation (SD). A paired *t*-test was used to assess statistical significance. All values remained within normal clinical reference ranges. *p* < 0.05 was considered statistically significant.

**Table 6 pharmaceutics-17-00880-t006:** Summary of particle size distribution and size span of the two formulations.

	STD	LMM
D_10%_ (μm)	17.0	212
D_50%_ (μm)	98.1	255
D_90%_ (μm)	302	297
Size Span	2.90	0.335

D_N%_ reflects the percentage of particles (N%) with a diameter (D) less than or equal to the specified value. The size span, calculated as (D90%–D10%)/D50%, reflects the uniformity of the particle distribution, with lower values indicating more monodisperse systems. The LMM formulation exhibited a narrower size span and larger median particle size, consistent with more uniform colloidal dispersion.

## Data Availability

The original contributions presented in this study are included in the article/Appendix A. Further inquiries can be directed to the corresponding author.

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
