# Peer review of "Novel Micellar Formulation of Silymarin (Milk Thistle) with Enhanced Bioavailability in a Double-Blind, Randomized, Crossover Human Trial"

_pharmaceutics, 2025, doi:10.3390/pharmaceutics17070880_

Round 1
Reviewer 1 Report (Previous Reviewer 1)
Comments and Suggestions for Authors
The authors evaluated the pharmacokinetics of a micellar milk thistle formulation of silymarin, compared to an unformulated milk thistle product, in a small-scale human bioavailability trial.
The study is interesting, but the results need to be discussed more critically.
My recommendation is major revisions.
My comments are:
Improve figure 2, the text in the pdf file is not readable
Micelles are colloidal aggregates with dimensions in the order of nanometers. The micelle sizes reported in the study are 10–80 μm defined by cryo-TEM and 255 μm by DLS. Why do the authors talk about micelles in the text and how do they explain these very different values?
In figure 6: the hydrodynamic particle size distribution of STD makes little sense. It is a commercial powder, the analysis should be done on the dry sample
Please specify in the caption of table 3 the acronyms used in the table
Authors should add in the text a comment on the different pharmacokinetic behavior of STD and LMM taking into account the different excipients of the formulations.
Author Response
Please see attachment.

Reviewer 2 Report (Previous Reviewer 2)
Comments and Suggestions for Authors
- The language used in the manuscript is mostly clear and the research is well-documented. However, there are a few areas where the expression could be more concise and precise.
- The randomization and blinding process could be described in more detail. For example, how was the allocation sequence generated and maintained to prevent selection bias?
- When explaining the use of paired t-tests with Holm correction, a brief justification of why this particular correction method was selected would be helpful for readers who may not be familiar with it.
- The discussion could delve deeper into the potential clinical implications of these results
- The manuscript states that no adverse events were observed in either group. However, the methods for monitoring adverse events could be described in more detail. How were adverse events assessed and recorded? Were there any predefined criteria for what constituted an adverse event? Providing this information would give readers a better understanding of the safety profile of the formulations.
- How do the newly observed cro-SEM differences in particle size and morphology relate to the improved bioavailability of the micellar formulation?
- How the limitations mentioned in this study might have affected the study outcomes? How to suggest possible ways to address them in future research?
Round 2
Reviewer 1 Report (Previous Reviewer 1)
Comments and Suggestions for Authors
The authors have modified the manuscript in accordance with the reviewer's comments.
Author Response
We thank the reviewer for their time and effort in reviewing our manuscript.
Reviewer 2 Report (Previous Reviewer 2)
Comments and Suggestions for Authors
The authors have replied all comments. The schemes, figs in the text should be further improved for better expression.
Author Response
Please see the attachment.

This manuscript is a resubmission of an earlier submission. The following is a list of the peer review reports and author responses from that submission.
Round 1
Reviewer 1 Report
Comments and Suggestions for Authors
The manuscript reports a pharmacokinetic study of a micellar formulation of milk thistle designed to improve the absorption of silymarin, compared to a non-formulated milk thistle, in a human bioavailability study. The study is interesting and shows encouraging results in favor of the formulation. However, the manuscript completely lacks the characterization of the formulation from a technological point of view: the components, the determination of parameters such as size, PdI, zeta potential, encapsulation efficiency, the in vitro release study, the stability evaluation.
- It may be interesting to compare the formulation with the extract alone, but the authors do not provide any information about the formulation.
- There are already many studies on silymarin, including clinical studies, however, the new formulation could represent a further possibility to improve bioavailability of silymarin
- Chemical-physical characterization of micelles
- The formulation data is missing. Results are based on clinical data only
Without this part I believe that the manuscript cannot be accepted for publication.
Reviewer 2 Report
Comments and Suggestions for Authors
1. The study enrolled 16 participants, which seems inadequate to ensure statistical power, especially given the high inter-subject variability.
2. The use of hard-gel vs. soft-gel capsules introduces a critical flaw in blinding. Participants could easily distinguish treatments, introducing bias in subjective outcomes, despite claims of "double-blinding."
3. The micellar formulation includes excipients like medium-chain triglycerides and phosphatidylcholine, which may independently enhance absorption. No control group with excipients alone was included, making it impossible to isolate the micellar system's contribution.
4. The elimination half-life and clearance mechanisms were inadequately discussed. The non-significant T1/2T1/2 difference (p=0.195p=0.195) contradicts the claimed "faster systemic clearance" for LMM, yet this contradiction is glossed over.
5. The 11.4-fold AUC increase is compared to an unformulated product but lacks direct comparison to established formulations like phytosomes. This omission misleads readers about LMM's novelty relative to existing technologies.
6. The use of geometric means without justification for log-normal distribution in small-sample data (n=16) risks overestimating effects. Holm-Sidak correction is mentioned but not transparently applied to all parameters, raising concerns about Type I error inflation.
7. Adverse event monitoring relied solely on self-reported questionnaires without objective measures like liver/kidney function tests. A 24-hour observation period is insufficient to evaluate hepatoprotective agents’ safety profiles.
8. The ClinicalTrials.gov registration number NCT06882681 is listed but not hyperlinked or validated. No IRB approval date or detailed consent documentation is provided, violating CONSORT guidelines.
9. The longer MRT for STD is attributed to "delayed absorption" without mechanistic evidence like solubility or enzymatic degradation. This speculative conclusion lacks support from in vitro dissolution or permeability assays.
10. References include non-peer-reviewed preprints, weakening the scientific foundation of the discussion.
11. The 130 mg dose was selected without rationale like alignment with prior clinical studies or dose-response data. This arbitrary choice casts doubt on the clinical relevance of the findings.
12. LC-HRMS parameters like ionization mode, mass accuracy are omitted.
